# Algorithms for CVaR Optimization in MDPs

**Yinlam Chow**[*]
Institute of Computational & Mathematical Engineering, Stanford University

**Mohammad Ghavamzadeh**[†]
Adobe Research & INRIA Lille - Team SequeL

## Abstract

In many sequential decision-making problems we may want to manage risk by minimizing some measure of variability in costs in addition to minimizing a standard criterion. Conditional value-at-risk (CVaR) is a relatively new risk measure that addresses some of the shortcomings of the well-known variance-related risk measures, and because of its computational efficiencies has gained popularity in finance and operations research. In this paper, we consider the mean-CVaR optimization problem in MDPs. We first derive a formula for computing the gradient of this risk-sensitive objective function. We then devise policy gradient and actor-critic algorithms that each uses a specific method to estimate this gradient and updates the policy parameters in the descent direction. We establish the convergence of our algorithms to locally risk-sensitive optimal policies. Finally, we demonstrate the usefulness of our algorithms in an optimal stopping problem.

## 1   Introduction

A standard optimization criterion for an infinite horizon Markov decision process (MDP) is the *expected sum of (discounted) costs* (i.e., finding a policy that minimizes the value function of the initial state of the system). However in many applications, we may prefer to minimize some measure of *risk* in addition to this standard optimization criterion. In such cases, we would like to use a criterion that incorporates a penalty for the *variability* (due to the stochastic nature of the system) induced by a given policy. In *risk-sensitive* MDPs [16], the objective is to minimize a risk-sensitive criterion such as the expected exponential utility [16], a variance-related measure [24, 14], or the percentile performance [15]. The issue of how to construct such criteria in a manner that will be both conceptually meaningful and mathematically tractable is still an open question.

Although most losses (returns) are not normally distributed, the typical Markowitz mean-variance optimization [18], that relies on the first two moments of the loss (return) distribution, has dominated the risk management for over 50 years. Numerous alternatives to mean-variance optimization have emerged in the literature, but there is no clear leader amongst these alternative risk-sensitive objective functions. *Value-at-risk* (VaR) and *conditional value-at-risk* (CVaR) are two promising such alternatives that quantify the losses that might be encountered in the tail of the loss distribution, and thus, have received high status in risk management. For (continuous) loss distributions, while $\text{VaR}_\alpha$ measures risk as the maximum loss that might be incurred w.r.t. a given confidence level $\alpha$, $\text{CVaR}_\alpha$ measures it as the expected loss given that the loss is greater or equal to $\text{VaR}_\alpha$. Although VaR is a popular risk measure, CVaR's computational advantages over VaR has boosted the development of CVaR optimization techniques. We provide the exact definitions of these two risk measures and briefly discuss some of the VaR's shortcomings in Section 2. CVaR minimization was first developed by Rockafellar and Uryasev [23] and its numerical effectiveness was demonstrated in portfolio optimization and option hedging problems. Their work was then extended to objective functions consist of different combinations of the expected loss and the CVaR, such as the minimization of the expected loss subject to a constraint on CVaR. This is the objective function

---

[*]Part of the work is completed during Yinlam Chow's internship at Adobe Research.
[†]Mohammad Ghavamzadeh is at Adobe Research, on leave of absence from INRIA Lille - Team SequeL.

that we study in this paper, although we believe that our proposed algorithms can be easily extended to several other CVaR-related objective functions. Boda and Filar [9] and Bäuerle and Ott [20, 3] extended the results of [23] to MDPs (sequential decision-making). While the former proposed to use dynamic programming (DP) to optimize CVaR, an approach that is limited to small problems, the latter showed that in both finite and infinite horizon MDPs, there exists a *deterministic history-dependent* optimal policy for CVaR optimization (see Section 3 for more details).

Most of the work in risk-sensitive sequential decision-making has been in the context of MDPs (when the model is known) and much less work has been done within the reinforcement learning (RL) framework. In risk-sensitive RL, we can mention the work by Borkar [10, 11] who considered the expected exponential utility and those by Tamar et al. [26] and Prashanth and Ghavamzadeh [17] on several variance-related risk measures. CVaR optimization in RL is a rather novel subject. Morimura et al. [19] estimate the return distribution while exploring using a CVaR-based risk-sensitive policy. Their algorithm does not scale to large problems. Petrik and Subramanian [22] propose a method based on stochastic dual DP to optimize CVaR in large-scale MDPs. However, their method is limited to linearly controllable problems. Borkar and Jain [12] consider a finite-horizon MDP with CVaR constraint and sketch a stochastic approximation algorithm to solve it. Finally, Tamar et al. [27] have recently proposed a policy gradient algorithm for CVaR optimization.

In this paper, we develop policy gradient (PG) and actor-critic (AC) algorithms for mean-CVaR optimization in MDPs. We first derive a formula for computing the gradient of this risk-sensitive objective function. We then propose several methods to estimate this gradient both incrementally and using system trajectories (update at each time-step vs. update after observing one or more trajectories). We then use these gradient estimations to devise PG and AC algorithms that update the policy parameters in the descent direction. Using the ordinary differential equations (ODE) approach, we establish the asymptotic convergence of our algorithms to locally risk-sensitive optimal policies. Finally, we demonstrate the usefulness of our algorithms in an optimal stopping problem. In comparison to [27], while they develop a PG algorithm for CVaR optimization in stochastic shortest path problems that only considers continuous loss distributions, uses a biased estimator for VaR, is not incremental, and has no comprehensive convergence proof, here we study mean-CVaR optimization, consider both discrete and continuous loss distributions, devise both PG and (several) AC algorithms (trajectory-based and incremental – plus AC helps in reducing the variance of PG algorithms), and establish convergence proof for our algorithms.

## 2 Preliminaries

We consider problems in which the agent's interaction with the environment is modeled as a MDP. A MDP is a tuple $\mathcal{M} = (\mathcal{X}, \mathcal{A}, C, P, P_0)$, where $\mathcal{X} = \{1, \ldots, n\}$ and $\mathcal{A} = \{1, \ldots, m\}$ are the state and action spaces; $C(x, a) \in [-C_{\max}, C_{\max}]$ is the bounded cost random variable whose expectation is denoted by $c(x, a) = \mathbb{E}\big[C(x, a)\big]$; $P(\cdot|x, a)$ is the transition probability distribution; and $P_0(\cdot)$ is the initial state distribution. For simplicity, we assume that the system has a single initial state $x^0$, i.e., $P_0(x) = \mathbf{1}\{x = x^0\}$. All the results of the paper can be easily extended to the case that the system has more than one initial state. We also need to specify the rule according to which the agent selects actions at each state. A *stationary policy* $\mu(\cdot|x)$ is a probability distribution over actions, conditioned on the current state. In policy gradient and actor-critic methods, we define a class of parameterized stochastic policies $\big\{\mu(\cdot|x; \theta), x \in \mathcal{X}, \theta \in \Theta \subseteq R^{\kappa_1}\big\}$, estimate the gradient of a performance measure w.r.t. the policy parameters $\theta$ from the observed system trajectories, and then improve the policy by adjusting its parameters in the direction of the gradient. Since in this setting a policy $\mu$ is represented by its $\kappa_1$-dimensional parameter vector $\theta$, policy dependent functions can be written as a function of $\theta$ in place of $\mu$. So, we use $\mu$ and $\theta$ interchangeably in the paper. We denote by $d^\mu_\gamma(x|x^0) = (1 - \gamma) \sum_{k=0}^{\infty} \gamma^k \mathbb{P}(x_k = x|x_0 = x^0; \mu)$ and $\pi^\mu_\gamma(x, a|x^0) = d^\mu_\gamma(x|x^0)\mu(a|x)$ the $\gamma$-discounted visiting distribution of state $x$ and state-action pair $(x, a)$ under policy $\mu$, respectively.

Let $Z$ be a bounded-mean random variable, i.e., $\mathbb{E}[|Z|] < \infty$, with the cumulative distribution function $F(z) = \mathbb{P}(Z \leq z)$ (e.g., one may think of $Z$ as the loss of an investment strategy $\mu$). We define the *value-at-risk* at the confidence level $\alpha \in (0, 1)$ as $\text{VaR}_\alpha(Z) = \min\big\{z \mid F(z) \geq \alpha\big\}$. Here the minimum is attained because $F$ is non-decreasing and right-continuous in $z$. When $F$ is continuous and strictly increasing, $\text{VaR}_\alpha(Z)$ is the unique $z$ satisfying $F(z) = \alpha$, otherwise, the VaR equation can have no solution or a whole range of solutions. Although VaR is a popular risk measure, it suffers from being unstable and difficult to work with numerically when $Z$ is not

normally distributed, which is often the case as loss distributions tend to exhibit fat tails or empirical discreteness. Moreover, VaR is not a *coherent* risk measure [1] and more importantly does not quantify the losses that might be suffered beyond its value at the $\alpha$-tail of the distribution [23]. An alternative measure that addresses most of the VaR's shortcomings is *conditional value-at-risk*, $\mathrm{CVAR}_\alpha(Z)$, which is the mean of the $\alpha$-tail distribution of $Z$. If there is no probability atom at $\mathrm{VaR}_\alpha(Z)$, $\mathrm{CVaR}_\alpha(Z)$ has a unique value that is defined as $\mathrm{CVaR}_\alpha(Z) = \mathbb{E}\big[Z \mid Z \geq \mathrm{VaR}_\alpha(Z)\big]$. Rockafellar and Uryasev [23] showed that

$$\mathrm{CVaR}_\alpha(Z) = \min_{\nu \in \mathbb{R}} H_\alpha(Z, \nu) \triangleq \min_{\nu \in \mathbb{R}} \left\{ \nu + \frac{1}{1-\alpha} \mathbb{E}\big[(Z-\nu)^+\big] \right\}. \tag{1}$$

where $(x)^+ = \max(x, 0)$ represents the positive part of $x$. Note that as a function of $\nu$, $H_\alpha(\cdot, \nu)$ is finite and convex (hence continuous).

## 3   CVaR Optimization in MDPs

For a policy $\mu$, we define the loss of a state $x$ (state-action pair $(x, a)$) as the sum of (discounted) costs encountered by the agent when it starts at state $x$ (state-action pair $(x, a)$) and then follows policy $\mu$, i.e., $D^\theta(x) = \sum_{k=0}^\infty \gamma^k C(x_k, a_k) \mid x_0 = x, \mu$ and $D^\theta(x, a) = \sum_{k=0}^\infty \gamma^k C(x_k, a_k) \mid x_0 = x, a_0 = a, \mu$. The expected value of these two random variables are the value and action-value functions of policy $\mu$, i.e., $V^\theta(x) = \mathbb{E}\big[D^\theta(x)\big]$ and $Q^\theta(x, a) = \mathbb{E}\big[D^\theta(x, a)\big]$. The goal in the standard discounted formulation is to find an optimal policy $\theta^* = \mathrm{argmin}_\theta V^\theta(x^0)$.

For CVaR optimization in MDPs, we consider the following optimization problem: For a given confidence level $\alpha \in (0, 1)$ and loss tolerance $\beta \in \mathbb{R}$,

$$\min_\theta V^\theta(x^0) \qquad \text{subject to} \qquad \mathrm{CVaR}_\alpha\big(D^\theta(x^0)\big) \leq \beta. \tag{2}$$

By Theorem 16 in [23], the optimization problem (2) is equivalent to ($H_\alpha$ is defined by (1))

$$\min_{\theta, \nu} V^\theta(x^0) \qquad \text{subject to} \qquad H_\alpha\big(D^\theta(x^0), \nu\big) \leq \beta. \tag{3}$$

To solve (3), we employ the Lagrangian relaxation procedure [4] to convert it to the following unconstrained problem:

$$\max_{\lambda \geq 0} \min_{\theta, \nu} \left( L(\theta, \nu, \lambda) \triangleq V^\theta(x^0) + \lambda\Big(H_\alpha\big(D^\theta(x^0), \nu\big) - \beta\Big) \right), \tag{4}$$

where $\lambda$ is the Lagrange multiplier. The goal here is to find the saddle point of $L(\theta, \nu, \lambda)$, i.e., a point $(\theta^*, \nu^*, \lambda^*)$ that satisfies $L(\theta, \nu, \lambda^*) \geq L(\theta^*, \nu^*, \lambda^*) \geq L(\theta^*, \nu^*, \lambda), \forall \theta, \nu, \forall \lambda \geq 0$. This is achieved by descending in $(\theta, \nu)$ and ascending in $\lambda$ using the gradients of $L(\theta, \nu, \lambda)$ w.r.t. $\theta$, $\nu$, and $\lambda$, i.e.,[1]

$$\nabla_\theta L(\theta, \nu, \lambda) = \nabla_\theta V^\theta(x^0) + \frac{\lambda}{(1-\alpha)} \nabla_\theta \mathbb{E}\Big[\big(D^\theta(x^0) - \nu\big)^+\Big], \tag{5}$$

$$\partial_\nu L(\theta, \nu, \lambda) = \lambda\left(1 + \frac{1}{(1-\alpha)} \partial_\nu \mathbb{E}\Big[\big(D^\theta(x^0) - \nu\big)^+\Big]\right) \ni \lambda\left(1 - \frac{1}{(1-\alpha)} \mathbb{P}\big(D^\theta(x^0) \geq \nu\big)\right), \tag{6}$$

$$\nabla_\lambda L(\theta, \nu, \lambda) = \nu + \frac{1}{(1-\alpha)} \mathbb{E}\Big[\big(D^\theta(x^0) - \nu\big)^+\Big] - \beta. \tag{7}$$

We assume that there exists a policy $\mu(\cdot|\cdot; \theta)$ such that $\mathrm{CVaR}_\alpha\big(D^\theta(x^0)\big) \leq \beta$ (feasibility assumption). As discussed in Section 1, Bäuerle and Ott [20, 3] showed that there exists a *deterministic history-dependent* optimal policy for CVaR optimization. The important point is that this policy does not depend on the complete history, but only on the current time step $k$, current state of the system $x_k$, and accumulated discounted cost $\sum_{i=0}^k \gamma^i C(x_i, a_i)$.

In the following, we present a policy gradient (PG) algorithm (Sec. 4) and several actor-critic (AC) algorithms (Sec. 5) to optimize (4). While the PG algorithm updates its parameters after observing several trajectories, the AC algorithms are incremental and update their parameters at each time-step.

## 4 A Trajectory-based Policy Gradient Algorithm

In this section, we present a policy gradient algorithm to solve the optimization problem (4). The unit of observation in this algorithm is a system trajectory generated by following the current policy. At each iteration, the algorithm generates $N$ trajectories by following the current policy, use them to estimate the gradients in Eqs. 5-7, and then use these estimates to update the parameters $\theta, \nu, \lambda$.

Let $\xi = \{x_0, a_0, x_1, a_1, \ldots, x_{T-1}, a_{T-1}, x_T\}$ be a trajectory generated by following the policy $\theta$, where $x_0 = x^0$ and $x_T$ is usually a terminal state of the system. After $x_k$ visits the terminal state, it enters a recurring sink state $x_S$ at the next time step, incurring zero cost, i.e., $C(x_S, a) = 0$, $\forall a \in \mathcal{A}$. Time index $T$ is referred to as the stopping time of the MDP. Since the transition is stochastic, $T$ is a non-deterministic quantity. Here we assume that the policy $\mu$ is proper, i.e., $\sum_{k=0}^{\infty} \mathbb{P}(x_k = x | x_0 = x^0, \mu) < \infty$ for every $x \notin \{x_S\}$. This further means that with probability 1, the MDP exits the transient states and hits $x_S$ (and stays in $x_S$) in finite time $T$. For simplicity, we assume that the agent incurs zero cost at the terminal state. Analogous results for the general case with a non-zero terminal cost can be derived using identical arguments. The loss and probability of $\xi$ are defined as $D(\xi) = \sum_{k=0}^{T-1} \gamma^k c(x_k, a_k)$ and $\mathbb{P}_\theta(\xi) = P_0(x_0) \prod_{k=0}^{T-1} \mu(a_k | x_k; \theta) P(x_{k+1} | x_k, a_k)$, respectively. It can be easily shown that $\nabla_\theta \log \mathbb{P}_\theta(\xi) = \sum_{k=0}^{T-1} \nabla_\theta \log \mu(a_k | x_k; \theta)$.

Algorithm 1 contains the pseudo-code of our proposed policy gradient algorithm. What appears inside the parentheses on the right-hand-side of the update equations are the estimates of the gradients of $L(\theta, \nu, \lambda)$ w.r.t. $\theta, \nu, \lambda$ (estimates of Eqs. 5-7) (see Appendix A.2 of [13]). $\Gamma_\theta$ is an operator that projects a vector $\theta \in \mathbb{R}^{\kappa_1}$ to the closest point in a compact and convex set $\Theta \subset \mathbb{R}^{\kappa_1}$, and $\Gamma_\nu$ and $\Gamma_\lambda$ are projection operators to $[-\frac{C_{\max}}{1-\gamma}, \frac{C_{\max}}{1-\gamma}]$ and $[0, \lambda_{\max}]$, respectively. These projection operators are necessary to ensure the convergence of the algorithm. The step-size schedules satisfy the standard conditions for stochastic approximation algorithms, and ensure that the VaR parameter $\nu$ update is on the fastest time-scale $\{\zeta_3(i)\}$, the policy parameter $\theta$ update is on the intermediate time-scale $\{\zeta_2(i)\}$, and the Lagrange multiplier $\lambda$ update is on the slowest time-scale $\{\zeta_1(i)\}$ (see Appendix A.1 of [13] for the conditions on the step-size schedules). This results in a three time-scale stochastic approximation algorithm. We prove that our policy gradient algorithm converges to a (local) saddle point of the risk-sensitive objective function $L(\theta, \nu, \lambda)$ (see Appendix A.3 of [13]).

---

**Algorithm 1** Trajectory-based Policy Gradient Algorithm for CVaR Optimization

---

**Input:** parameterized policy $\mu(\cdot | \cdot; \theta)$, confidence level $\alpha$, and loss tolerance $\beta$
**Initialization:** policy parameter $\theta = \theta_0$, VaR parameter $\nu = \nu_0$, and the Lagrangian parameter $\lambda = \lambda_0$
  **for** $i = 0, 1, 2, \ldots$ **do**
    **for** $j = 1, 2, \ldots$ **do**
      Generate $N$ trajectories $\{\xi_{j,i}\}_{j=1}^N$ by starting at $x_0 = x^0$ and following the current policy $\theta_i$.
    **end for**

$\nu$ **Update:** $\quad \nu_{i+1} = \Gamma_\nu \left[ \nu_i - \zeta_3(i) \left( \lambda_i - \frac{\lambda_i}{(1-\alpha)N} \sum_{j=1}^N \mathbf{1}\{D(\xi_{j,i}) \geq \nu_i\} \right) \right]$

$\theta$ **Update:** $\quad \theta_{i+1} = \Gamma_\theta \left[ \theta_i - \zeta_2(i) \left( \frac{1}{N} \sum_{j=1}^N \nabla_\theta \log \mathbb{P}_\theta(\xi_{j,i}) |_{\theta=\theta_i} D(\xi_{j,i}) \right. \right.$

$\qquad\qquad\qquad\qquad \left. \left. + \frac{\lambda_i}{(1-\alpha)N} \sum_{j=1}^N \nabla_\theta \log \mathbb{P}_\theta(\xi_{j,i})|_{\theta=\theta_i} \left( D(\xi_{j,i}) - \nu_i \right) \mathbf{1}\{D(\xi_{j,i}) \geq \nu_i\} \right) \right]$

$\lambda$ **Update:** $\quad \lambda_{i+1} = \Gamma_\lambda \left[ \lambda_i + \zeta_1(i) \left( \nu_i - \beta + \frac{1}{(1-\alpha)N} \sum_{j=1}^N \left( D(\xi_{j,i}) - \nu_i \right) \mathbf{1}\{D(\xi_{j,i}) \geq \nu_i\} \right) \right]$

  **end for**
  **return** parameters $\nu, \theta, \lambda$

---

## 5 Incremental Actor-Critic Algorithms

As mentioned in Section 4, the unit of observation in our policy gradient algorithm (Algorithm 1) is a system trajectory. This may result in high variance for the gradient estimates, especially when the length of the trajectories is long. To address this issue, in this section, we propose two actor-critic

algorithms that use linear approximation for some quantities in the gradient estimates and update the parameters incrementally (after each state-action transition). We present two actor-critic algorithms for optimizing the risk-sensitive measure (4). These algorithms are based on the gradient estimates of Sections 5.1-5.3. While the first algorithm (SPSA-based) is fully incremental and updates all the parameters $\theta, \nu, \lambda$ at each time-step, the second one updates $\theta$ at each time-step and updates $\nu$ and $\lambda$ only at the end of each trajectory, thus given the name semi trajectory-based. Algorithm 2 contains the pseudo-code of these algorithms. The projection operators $\Gamma_\theta$, $\Gamma_\nu$, and $\Gamma_\lambda$ are defined as in Section 4 and are necessary to ensure the convergence of the algorithms. The step-size schedules satisfy the standard conditions for stochastic approximation algorithms, and ensures that the critic update is on the fastest time-scale $\{\zeta_4(i)\}$, the policy and VaR parameter updates are on the intermediate time-scale, with $\nu$-update $\{\zeta_3(i)\}$ being faster than $\theta$-update $\{\zeta_2(i)\}$, and finally the Lagrange multiplier update is on the slowest time-scale $\{\zeta_1(i)\}$ (see Appendix B.1 of [13] for the conditions on these step-size schedules). This results in four time-scale stochastic approximation algorithms. We prove that these actor-critic algorithms converge to a (local) saddle point of the risk-sensitive objective function $L(\theta, \nu, \lambda)$ (see Appendix B.4 of [13]).

## 5.1 Gradient w.r.t. the Policy Parameters $\theta$

The gradient of our objective function w.r.t. the policy parameters $\theta$ in (5) may be rewritten as

$$\nabla_\theta L(\theta, \nu, \lambda) = \nabla_\theta \left( \mathbb{E}\big[D^\theta(x^0)\big] + \frac{\lambda}{(1-\alpha)} \mathbb{E}\Big[\big(D^\theta(x^0) - \nu\big)^+\Big] \right). \tag{8}$$

Given the original MDP $\mathcal{M} = (\mathcal{X}, \mathcal{A}, C, P, P_0)$ and the parameter $\lambda$, we define the augmented MDP $\bar{\mathcal{M}} = (\bar{\mathcal{X}}, \bar{\mathcal{A}}, \bar{C}, \bar{P}, \bar{P}_0)$ as $\bar{\mathcal{X}} = \mathcal{X} \times \mathbb{R}$, $\bar{\mathcal{A}} = \mathcal{A}$, $\bar{P}_0(x, s) = P_0(x)\mathbf{1}\{s_0 = s\}$, and

$$\bar{C}(x, s, a) = \begin{cases} \lambda(-s)^+/(1-\alpha) & \text{if } x = x_T \\ C(x, a) & \text{otherwise} \end{cases}, \bar{P}(x', s'|x, s, a) = \begin{cases} P(x'|x, a) & \text{if } s' = \big(s - C(x, a)\big)/\gamma \\ 0 & \text{otherwise} \end{cases}$$

where $x_T$ is any terminal state of the original MDP $\mathcal{M}$ and $s_T$ is the value of the $s$ part of the state when a policy $\theta$ reaches a terminal state $x_T$ after $T$ steps, i.e., $s_T = \frac{1}{\gamma^T}\big(\nu - \sum_{k=0}^{T-1} \gamma^k C(x_k, a_k)\big)$. We define a class of parameterized stochastic policies $\{\mu(\cdot|x, s; \theta), (x, s) \in \bar{\mathcal{X}}, \theta \in \Theta \subseteq R^{\kappa_1}\}$ for this augmented MDP. Thus, the total (discounted) loss of this trajectory can be written as

$$\sum_{k=0}^{T-1} \gamma^k C(x_k, a_k) + \gamma^T \bar{C}(x_T, s_T, a) = D^\theta(x^0) + \frac{\lambda}{(1-\alpha)}\big(D^\theta(x^0) - \nu\big)^+. \tag{9}$$

From (9), it is clear that the quantity in the parenthesis of (8) is the value function of the policy $\theta$ at state $(x^0, \nu)$ in the augmented MDP $\bar{\mathcal{M}}$, i.e., $V^\theta(x^0, \nu)$. Thus, it is easy to show that (the second equality in Eq. 10 is the result of the policy gradient theorem [21])

$$\nabla_\theta L(\theta, \nu, \lambda) = \nabla_\theta V^\theta(x^0, \nu) = \frac{1}{1-\gamma} \sum_{x,s,a} \pi_\gamma^\theta(x, s, a|x^0, \nu) \nabla \log \mu(a|x, s; \theta) Q^\theta(x, s, a), \tag{10}$$

where $\pi_\gamma^\theta$ is the discounted visiting distribution (defined in Section 2) and $Q^\theta$ is the action-value function of policy $\theta$ in the augmented MDP $\bar{\mathcal{M}}$. We can show that $\frac{1}{1-\gamma} \nabla \log \mu(a_k|x_k, s_k; \theta) \cdot \delta_k$ is an unbiased estimate of $\nabla_\theta L(\theta, \nu, \lambda)$, where $\delta_k = \bar{C}(x_k, s_k, a_k) + \gamma\widehat{V}(x_{k+1}, s_{k+1}) - \widehat{V}(x_k, s_k)$ is the temporal-difference (TD) error in $\bar{\mathcal{M}}$, and $\widehat{V}$ is an unbiased estimator of $V^\theta$ (see e.g., [6, 7]). In our actor-critic algorithms, the critic uses linear approximation for the value function $V^\theta(x, s) \approx v^\top \phi(x, s) = \widetilde{V}^{\theta, v}(x, s)$, where the feature vector $\phi(\cdot)$ belongs to the low-dimensional space $\mathbb{R}^{\kappa_2}$.

## 5.2 Gradient w.r.t. the Lagrangian Parameter $\lambda$

We may rewrite the gradient of our objective function w.r.t. the Lagrangian parameters $\lambda$ in (7) as

$$\nabla_\lambda L(\theta, \nu, \lambda) = \nu - \beta + \nabla_\lambda \left( \mathbb{E}\big[D^\theta(x^0)\big] + \frac{\lambda}{(1-\alpha)} \mathbb{E}\Big[\big(D^\theta(x^0) - \nu\big)^+\Big] \right) \overset{(a)}{=} \nu - \beta + \nabla_\lambda V^\theta(x^0, \nu). \tag{11}$$

Similar to Section 5.1, **(a)** comes from the fact that the quantity in the parenthesis in (11) is $V^\theta(x^0, \nu)$, the value function of the policy $\theta$ at state $(x^0, \nu)$ in the augmented MDP $\bar{\mathcal{M}}$. Note that the dependence of $V^\theta(x^0, \nu)$ on $\lambda$ comes from the definition of the cost function $\bar{C}$ in $\bar{\mathcal{M}}$. We now derive an expression for $\nabla_\lambda V^\theta(x^0, \nu)$, which in turn will give us an expression for $\nabla_\lambda L(\theta, \nu, \lambda)$.

**Lemma 1** *The gradient of $V^\theta(x^0, \nu)$ w.r.t. the Lagrangian parameter $\lambda$ may be written as*

$$\nabla_\lambda V^\theta(x^0, \nu) = \frac{1}{1-\gamma} \sum_{x,s,a} \pi_\gamma^\theta(x, s, a | x^0, \nu) \frac{1}{(1-\alpha)} \mathbf{1}\{x = x_T\}(-s)^+. \tag{12}$$

*Proof.* See Appendix B.2 of [13]. ∎

From Lemma 1 and (11), it is easy to see that $\nu - \beta + \frac{1}{(1-\gamma)(1-\alpha)} \mathbf{1}\{x = x_T\}(-s)^+$ is an unbiased estimate of $\nabla_\lambda L(\theta, \nu, \lambda)$. An issue with this estimator is that its value is fixed to $\nu_k - \beta$ all along a system trajectory, and only changes at the end to $\nu_k - \beta + \frac{1}{(1-\gamma)(1-\alpha)}(-s_T)^+$. This may affect the incremental nature of our actor-critic algorithm. To address this issue, we propose a different approach to estimate the gradients w.r.t. $\theta$ and $\lambda$ in Sec. 5.4 (of course this does not come for free).

Another important issue is that the above estimator is unbiased only if the samples are generated from the distribution $\pi_\gamma^\theta(\cdot | x^0, \nu)$. If we just follow the policy, then we may use $\nu_k - \beta + \frac{\gamma^k}{(1-\alpha)} \mathbf{1}\{x_k = x_T\}(-s_k)^+$ as an estimate for $\nabla_\lambda L(\theta, \nu, \lambda)$. Note that this is an issue for all discounted actor-critic algorithms that their (likelihood ratio based) estimate for the gradient is unbiased only if the samples are generated from $\pi_\gamma^\theta$, and not when we simply follow the policy. This might be a reason that we have no convergence analysis (to the best of our knowledge) for (likelihood ratio based) discounted actor-critic algorithms.[2]

### 5.3 Sub-Gradient w.r.t. the VaR Parameter $\nu$

We may rewrite the sub-gradient of our objective function w.r.t. the VaR parameter $\nu$ (Eq. 6) as

$$\partial_\nu L(\theta, \nu, \lambda) \ni \lambda\left(1 - \frac{1}{(1-\alpha)}\mathbb{P}\Big(\sum_{k=0}^{\infty} \gamma^k C(x_k, a_k) \geq \nu \mid x_0 = x^0; \theta\Big)\right). \tag{13}$$

From the definition of the augmented MDP $\bar{\mathcal{M}}$, the probability in (13) may be written as $\mathbb{P}(s_T \leq 0 \mid x_0 = x^0, s_0 = \nu; \theta)$, where $s_T$ is the $s$ part of the state in $\bar{\mathcal{M}}$ when we reach a terminal state, i.e., $x = x_T$ (see Section 5.1). Thus, we may rewrite (13) as

$$\partial_\nu L(\theta, \nu, \lambda) \ni \lambda\left(1 - \frac{1}{(1-\alpha)}\mathbb{P}\big(s_T \leq 0 \mid x_0 = x^0, s_0 = \nu; \theta\big)\right). \tag{14}$$

From (14), it is easy to see that $\lambda - \lambda\mathbf{1}\{s_T \leq 0\}/(1-\alpha)$ is an unbiased estimate of the sub-gradient of $L(\theta, \nu, \lambda)$ w.r.t. $\nu$. An issue with this (unbiased) estimator is that it can be only applied at the end of a system trajectory (i.e., when we reach the terminal state $x_T$), and thus, using it prevents us of having a fully incremental algorithm. In fact, this is the estimator that we use in our *semi trajectory-based* actor-critic algorithm.

One approach to estimate this sub-gradient incrementally is to use *simultaneous perturbation stochastic approximation* (SPSA) method [8]. The idea of SPSA is to estimate the sub-gradient $g(\nu) \in \partial_\nu L(\theta, \nu, \lambda)$ using two values of $g$ at $\nu^- = \nu - \Delta$ and $\nu^+ = \nu + \Delta$, where $\Delta > 0$ is a positive perturbation (see [8, 17] for the detailed description of $\Delta$).[3] In order to see how SPSA can help us to estimate our sub-gradient incrementally, note that

$$\partial_\nu L(\theta, \nu, \lambda) = \lambda + \partial_\nu \left(\mathbb{E}\big[D^\theta(x^0)\big] + \frac{\lambda}{(1-\alpha)}\mathbb{E}\big[\big(D^\theta(x^0) - \nu\big)^+\big]\right) \overset{(a)}{=} \lambda + \partial_\nu V^\theta(x^0, \nu). \tag{15}$$

Similar to Sections 5.1, **(a)** comes from the fact that the quantity in the parenthesis in (15) is $V^\theta(x^0, \nu)$, the value function of the policy $\theta$ at state $(x^0, \nu)$ in the augmented MDP $\bar{\mathcal{M}}$. Since the critic uses a linear approximation for the value function, i.e., $V^\theta(x, s) \approx v^\top\phi(x, s)$, in our actor-critic algorithms (see Section 5.1 and Algorithm 2), the SPSA estimate of the sub-gradient would be of the form $g(\nu) \approx \lambda + v^\top\big[\phi(x^0, \nu^+) - \phi(x^0, \nu^-)\big]/2\Delta$.

### 5.4 An Alternative Approach to Compute the Gradients

In this section, we present an alternative way to compute the gradients, especially those w.r.t. $\theta$ and $\lambda$. This allows us to estimate the gradient w.r.t. $\lambda$ in a (more) incremental fashion (compared to the method of Section 5.3), with the cost of the need to use two different linear function approximators

(instead of one used in Algorithm 2). In this approach, we define the augmented MDP slightly different than the one in Section 5.3. The only difference is in the definition of the cost function, which is defined here as (note that $C(x, a)$ has been replaced by 0 and $\lambda$ has been removed)

$$\bar{C}(x, s, a) = \begin{cases} (-s)^+/(1 - \alpha) & \text{if } x = x_T, \\ 0 & \text{otherwise,} \end{cases}$$

where $x_T$ is any terminal state of the original MDP $\mathcal{M}$. It is easy to see that he term $\frac{1}{(1-\alpha)}\mathbb{E}\left[\left(D^\theta(x^0) - \nu\right)^+\right]$ appearing in the gradients of Eqs. 5-7 is the value function of the policy $\theta$ at state $(x^0, \nu)$ in this augmented MDP. As a result, we have

**Gradient w.r.t. $\theta$:** It is easy to see that now this gradient (Eq. 5) is the gradient of the value function of the original MDP, $\nabla_\theta V^\theta(x^0)$, plus $\lambda$ times the gradient of the value function of the augmented MDP, $\nabla_\theta V^\theta(x^0, \nu)$, both at the initial states of these MDPs (with abuse of notation, we use $V$ for the value function of both MDPs). Thus, using linear approximators $u^\top f(x, s)$ and $v^\top \phi(x, s)$ for the value functions of the original and augmented MDPs, $\nabla_\theta L(\theta, \nu, \lambda)$ can be estimated as $\nabla_\theta \log \mu(a_k|x_k, s_k; \theta) \cdot (\epsilon_k + \lambda\delta_k)$, where $\epsilon_k$ and $\delta_k$ are the TD-errors of these MDPs.

**Gradient w.r.t. $\lambda$:** Similar to the case for $\theta$, it is easy to see that this gradient (Eq. 7) is $\nu - \beta$ plus the value function of the augmented MDP, $V^\theta(x^0, \nu)$, and thus, can be estimated *incrementally* as $\nabla_\lambda L(\theta, \nu, \lambda) \approx \nu - \beta + v^\top \phi(x, s)$.

**Sub-Gradient w.r.t. $\nu$:** This sub-gradient (Eq. 6) is $\lambda$ times one plus the gradient w.r.t. $\nu$ of the value function of the augmented MDP, $\nabla_\nu V^\theta(x^0, \nu)$, and thus, it can be estimated *incrementally* using SPSA as $\lambda\left(1 + \frac{v^\top\left[\phi(x^0, \nu^+) - \phi(x^0, \nu^-)\right]}{2\Delta}\right)$.

Algorithm 3 in Appendix B.3 of [13] contains the pseudo-code of the resulting algorithm.

---

**Algorithm 2** Actor-Critic Algorithms for CVaR Optimization

---

**Input:** Parameterized policy $\mu(\cdot|\cdot; \theta)$ and value function feature vector $\phi(\cdot)$ (both over the augmented MDP $\bar{\mathcal{M}}$), confidence level $\alpha$, and loss tolerance $\beta$
**Initialization:** policy parameters $\theta = \theta_0$; VaR parameter $\nu = \nu_0$; Lagrangian parameter $\lambda = \lambda_0$; value function weight vector $v = v_0$
**// (1) SPSA-based Algorithm:**
**for** $k = 0, 1, 2, \ldots$ **do**
    Draw action $a_k \sim \mu(\cdot|x_k, s_k; \theta_k)$;            Observe cost $\bar{C}(x_k, s_k, a_k)$ (with $\lambda = \lambda_k$);
    Observe next state $(x_{k+1}, s_{k+1}) \sim \bar{P}(\cdot|x_k, s_k, a_k)$;   *// note that* $s_{k+1} = (s_k - C(x_k, a_k))/\gamma$

$$\textbf{TD Error:} \quad \delta_k = \bar{C}(x_k, s_k, a_k) + \gamma v_k^\top \phi(x_{k+1}, s_{k+1}) - v_k^\top \phi(x_k, s_k) \tag{16}$$

$$\textbf{Critic Update:} \quad v_{k+1} = v_k + \zeta_4(k)\delta_k \phi(x_k, s_k) \tag{17}$$

$$\boldsymbol{\nu} \textbf{ Update:} \quad \nu_{k+1} = \Gamma_\nu\left(\nu_k - \zeta_3(k)\left(\lambda_k + \frac{v_k^\top\left[\phi(x^0, \nu_k + \Delta_k) - \phi(x^0, \nu_k - \Delta_k)\right]}{2\Delta_k}\right)\right) \tag{18}$$

$$\boldsymbol{\theta} \textbf{ Update:} \quad \theta_{k+1} = \Gamma_\theta\left(\theta_k - \frac{\zeta_2(k)}{1 - \gamma}\nabla_\theta \log \mu(a_k|x_k, s_k; \theta) \cdot \delta_k\right) \tag{19}$$

$$\boldsymbol{\lambda} \textbf{ Update:} \quad \lambda_{k+1} = \Gamma_\lambda\left(\lambda_k + \zeta_1(k)\left(\nu_k - \beta + \frac{1}{(1 - \alpha)(1 - \gamma)}\mathbf{1}\{x_k = x_T\}(-s_k)^+\right)\right) \tag{20}$$

    **if** $x_k = x_T$ (reach a terminal state), **then** set $(x_{k+1}, s_{k+1}) = (x^0, \nu_{k+1})$
**end for**
**// (2) Semi Trajectory-based Algorithm:**
**for** $k = 0, 1, 2, \ldots$ **do**
    **if** $x_k \neq x_T$ **then**
        Draw action $a_k \sim \mu(\cdot|x_k, s_k; \theta_k)$, observe cost $\bar{C}(x_k, s_k, a_k)$ (with $\lambda = \lambda_k$), and next state $(x_{k+1}, s_{k+1}) \sim \bar{P}(\cdot|x_k, s_k, a_k)$;   Update $(\delta_k, v_k, \theta_k, \lambda_k)$ using Eqs. 16, 17, 19, and 20
    **else**
        Update $(\delta_k, v_k, \theta_k, \lambda_k)$ using Eqs. 16, 17, 19, and 20;   Update $\nu$ as

$$\boldsymbol{\nu} \textbf{ Update:} \quad \nu_{k+1} = \Gamma_\nu\left(\nu_k - \zeta_3(k)\left(\lambda_k - \frac{\lambda_k}{1 - \alpha}\mathbf{1}\{s_T \leq 0\}\right)\right) \tag{21}$$

        Set $(x_{k+1}, s_{k+1}) = (x^0, \nu_{k+1})$
    **end if**
**end for**
**return** policy and value function parameters $\theta, \nu, \lambda, v$

---

# 6 Experimental Results

We consider an optimal stopping problem in which the state at each time step $k \le T$ consists of the cost $c_k$ and time $k$, i.e., $x = (c_k, k)$, where $T$ is the stopping time. The agent (buyer) should decide either to accept the present cost or wait. If she accepts or when $k = T$, the system reaches a terminal state and the cost $c_k$ is received, otherwise, she receives the cost $p_h$ and the new state is $(c_{k+1}, k+1)$, where $c_{k+1}$ is $f_u c_k$ w.p. $p$ and $f_d c_k$ w.p. $1 - p$ ($f_u > 1$ and $f_d < 1$ are constants). Moreover, there is a discounted factor $\gamma \in (0, 1)$ to account for the increase in the buyer's affordability. The problem has been described in more details in Appendix C of [13]. Note that if we change cost to reward and minimization to maximization, this is exactly the American option pricing problem, a standard testbed to evaluate risk-sensitive algorithms (e.g., [26]). Since the state space is continuous, finding an exact solution via DP is infeasible, and thus, it requires approximation and sampling techniques.

We compare the performance of our risk-sensitive policy gradient Algorithm 1 (PG-CVaR) and two actor-critic Algorithms 2 (AC-CVaR-SPSA, AC-CVaR-Semi-Traj) with their risk-neutral counterparts (PG and AC) (see Appendix C of [13] for the details of these experiments). Figure 1 shows the distribution of the discounted cumulative cost $D^\theta(x^0)$ for the policy $\theta$ learned by each of these algorithms. The results indicate that the risk-sensitive algorithms yield a higher expected loss, but less variance, compared to the risk-neutral methods. More precisely, the loss distributions of the risk-sensitive algorithms have lower right-tail than their risk-neutral counterparts. Table 1 summarizes the performance of these algorithms. The numbers reiterate what we concluded from Figure 1.

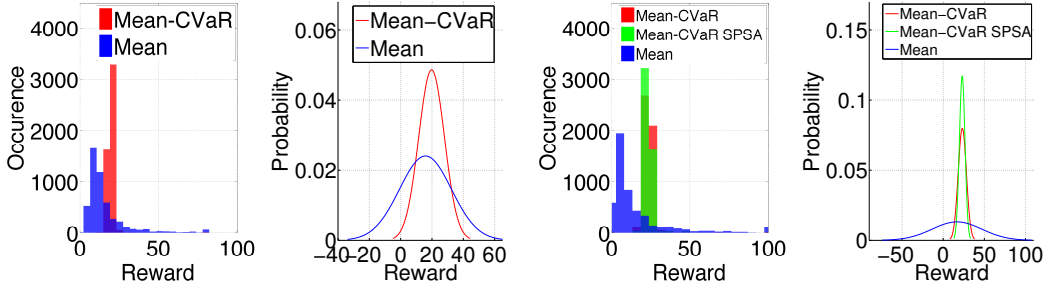

Figure 1: Loss distributions for the policies learned by the risk-sensitive and risk-neutral policy gradient and actor critic algorithms. The two left figures correspond to the PG methods, and the two right figures correspond to the AC algorithms. In all cases, the loss tolerance equals to $\beta = 40$.

|  | $\mathbb{E}(D^\theta(x^0))$ | $\sigma(D^\theta(x^0))$ | $\text{CVaR}(D^\theta(x^0))$ |
|---|---|---|---|
| PG | 16.08 | 17.53 | 69.18 |
| PG-CVaR | 19.75 | 7.06 | 25.75 |
| AC | 16.96 | 32.09 | 122.61 |
| AC-CVaR-SPSA | 22.86 | 3.40 | 31.36 |
| AC-CVaR-Semi-Traj. | 23.01 | 4.98 | 34.81 |

Table 1: Performance comparison for the policies learned by the risk-sensitive and risk-neutral algorithms.

# 7 Conclusions and Future Work

We proposed novel policy gradient and actor critic (AC) algorithms for CVaR optimization in MDPs. We provided proofs of convergence (in [13]) to locally risk-sensitive optimal policies for the proposed algorithms. Further, using an optimal stopping problem, we observed that our algorithms resulted in policies whose loss distributions have lower right-tail compared to their risk-neutral counterparts. This is extremely important for a risk averse decision-maker, especially if the right-tail contains catastrophic losses. Future work includes: **1)** Providing convergence proofs for our AC algorithms when the samples are generated by following the policy and not from its discounted visiting distribution, **2)** Using importance sampling methods [2, 27] to improve gradient estimates in the right-tail of the loss distribution (worst-case events that are observed with low probability) of the CVaR objective function, and **4)** Evaluating our algorithms in more challenging problems.

**Acknowledgement**    The authors would like to thank Professor Marco Pavone and Lucas Janson for their comments that helped us with some technical details in the proofs of the algorithms.

## Footnotes

[1] The notation $\ni$ in (6) means that the right-most term is a member of the sub-gradient set $\partial_\nu L(\theta, \nu, \lambda)$.

[2]Note that the discounted actor-critic algorithm with convergence proof in [5] is based on SPSA.

[3]SPSA-based gradient estimate was first proposed in [25] and has been widely used in various settings, especially those involving high-dimensional parameter. The SPSA estimate described above is two-sided. It can also be implemented single-sided, where we use the values of the function at $\nu$ and $\nu^+$. We refer the readers to [8] for more details on SPSA and to [17] for its application in learning in risk-sensitive MDPs.

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
