[Reviews · NeurIPS 2014]

Submitted by Assigned_Reviewer_2

This paper derives policy gradient algorithms for risk-sensitive MDPs for the particular criterion CVaR - a recent and popular criterion. First, the author derive gradients for the objective based on a Lagrangian relaxation of the constrained optimization. This naturally turns into a policy gradient algorithm where the expected return that appears in the gradient is estimated from full trajectories (reinforce-like). They then propose a scheme to obtain incremental actor-critic versions, where the critic computes the value (and other quantities) of an augmented MDP convenient for gradient estimation. An experimental section illustrates these algorithms on a small problem.

The paper is well-written, and the introduction clearly motivates the paper. The paper is quite technical, but I didn't have trouble understanding the notation and following the equations/algorithms. I am only mildly familiar with the literature on risk-sensitive MDPs so I can't provide insightful comments on the impact/significance of the contributions in the paper. My understanding is that, being incremental and compatible w/ function approximation, the algorithms in the paper have the potential to scale to larger domains than what is currently the norm with risk-sensitive planning.

A couple of questions:

- About the dependence of the policy on the accumulated cost and time, I understood that by making the feasibility assumption then you can restrict yourself to state-only (but stochastic) policies. Is that correct? Does it mean there might be better history-based policies that satisfy the CVaR constraint but that you are not considering? It might be worth clarifying this in the paper.

[EDIT: Thanks for clarifying]

- About the optimized policy, does converging to a saddle point in the Lagrangian necessarily imply that we reach any kind of local optimum in the original optimization problem? If not, then is it really fair to say that the algorithms converge to "locallly risk-sensitive optimal policies" (abstract)?

- In the intro, it is claimed that [27] (the closest competitor it seems) does not have a convergence proof for their policy-gradient algorithm, but a quick look at [27] reveals the following claim: "We [...] prove the convergence of the policy gradient algorithm to a local optimum". Maybe this is worth clarifying?

[EDIT: Fair enough, I didn't pay attention to the arxiv version/date of submission.]

While I appreciate the theoretical contribution, I was a bit disappointed by the experimental section, which does not really convey the impression of scalability and robustness (in contrast, [27] had experiments with the tetris domain using CVaR).

Future work #3 seem particularly critical for sample-based algorithms in this risk setting, so it is a pity that this wasn't addressed in this work.

Overall, a nice paper with clear contributions. Experiments are lacking but there is a lot of theoretical work.

Some minor typos:
- Equation 4: max lambda should have a non-negativity constraint(?)
- line 1313: xi_3 --> xi_4
- line 269 (footnote): SPSA not yet defined.

Note: I haven't verified all the proofs in the long supplementary material provided with the paper.
Summary: A clear paper that derives policy gradient (including actor critic) algorithms for a promising risk-sensitive MDP setting (CVaR), the main contributions are deriving the gradients, algorithms to compute them, and proving their theoretical properties. It's unfortunate that the experimental section is not enough to assess the practical impact of the algorithms, but the paper would likely benefit the community nonetheless.

Submitted by Assigned_Reviewer_22

This paper proposes policy gradient and actor-critic algorithms for minimizing the expected cumulative cost under the conditions that the conditional value at risk (CVaR) of cumulative cost is below a threshold. A straightforward approach (Algorithm 1) requires sampling many trajectories for estimating the gradients. The authors thus propose the approach of augmenting the state space with an estimate of value at risk, and approximating the value function with a linear function of features to deal with the continuous state space that results from the augmentation. Numerical experiments show that the proposed approach can adequately take risk into consideration.

This paper extends the recently proposed policy gradient algorithm of minimizing CVaR by Tamar et al. 2014 in several ways, but numerical experiments do not show comparison against Tamar et al. 2014. From algorithmic perspectives, the primary difference from Tamar et al. 2014 is in the use of augmented state space and approximation with linear function. I would like to understand how effective this approach is for example against the approach of importance sampling of Tamar et al. 2014.

Minor comments:

L56: Scalability is irrelevant here.

L95 and L97: The two phrases are conflicting: "the minimum is attained ..." and "VaR equation can have no solution or a whole range of solutions."

L206: Define s^0. Also how is s^0 different from s_0?

L338: How to initialize s_0?

====

In my original review, I was too much concerned with comparison against [27]. The authors are right in that [27] is not mature enough to be compared against in numerical experiments. It is too demanding to compare a proposed method against the paper that is revised after submission.

Without the necessity of the comparison against [27], I think the paper makes enough contributions as a NIPS paper. This is one of the first papers that propose policy gradient with the objective involving CVaR. The analysis appears to be sound. In particular, this paper deals with history-dependency appropriately, which was ignored in [27]. Numerical experiments are not very strong but demonstrate the advantages of the proposed approach.
Summary: This paper extends a recently proposed policy gradient algorithm of minimizing CVaR, but the advantage of the present work is unclear.

Submitted by Assigned_Reviewer_23

The paper describes how to extend two reinforcement learning algorithms---policy gradient and actor-critic---to risk-averse settings modeled using a CVaR constraint on the costs.

The paper is well written and addresses a generally important topic. However, the algorithms are all heuristics with no meaningful guarantees. The experimental results do not convincingly show that the proposed methods are any better than other baselines, and it is not clear that any kind of useful problems could be solved using the proposed methods.

The restriction to CVaR only is somewhat limiting, though it makes more sense in the constraints than in the objective function. It would be nice to see how to extent the method to any other convex/coherent risk measure.

The experimental results are quite limited. The difference between the solution quality for different algorithms is very small. Perhaps it would be helpful to add more discussion on the interpretation of the results. Also, results on a more realistic problem which actually requires risk aversion would be more convincing.

There have been a number of papers in other communities that solve multi-stage risk averse problems in domains raging from finance to resource management. Showing that the proposed methods can address such problems in better ways would make the paper much stronger.

The authors addressed my concerns in the rebuttal.
Summary: The paper addresses an important topic in a well-defined way, but comes up short in terms of evaluation (or theoretical contributions).
Author Feedback
Author rebuttal: We'd like to thank the reviewers for their useful comments. Before responding to each reviewer individually, we'd like to mention a few points that might be relevant to all the reviewers:

(a) The main focus of the paper is on developing policy gradient (PG) and actor-critic (AC) algos with convergence guarantees for mean-CVaR optimization. The (admittedly) simple experiments are only used as a proof of concept to show that the algos are successful in finding a policy that has a lighter tail than the one resulted from just optimizing the mean. Although we were careful to maintain a high-level of fairness in our comparison, we didn't spend much time to tune the parameters (policy and value function representations) to obtain the best possible result for each algo. We admit that evaluating the proposed algos on a more complex and meaningful (in terms of risk optimization) problem can heavily improve the quality of the paper.

(b) "comparison to the algo in [27] - the arxiv paper by Tamar et al."
The algo in [27] is PG that is similar to our proposed PG algo, which has been used in our experiments. To be more precise, the PG algo in [27] differs from ours in these ways:
1) it's for stochastic shortest path and not discounted MDPs
2) it optimizes CVaR and not mean-CVaR (it makes more sense, and to the best of our knowledge it is more common in practice, to optimize mean-CVaR and not just CVaR)
3) it uses a biased estimator for VaR instead of the way we update \nu.
Given all the above, our PG algo and the one in [27] are similar enough that it is not too unfair to claim that in fact we have evaluated the algo in [27] in our paper. Moreover, we recently compared our PG algo with the extension of Tamar's to mean-CVaR optimization (basically the only difference is the way the \nu parameter is updated) in our simple problem, and our algo performed better in terms of CVaR.

Rev2

"history-based policies"
It's known [3,19] that the optimal policy of the mean-CVaR optimization problem is history-dependent, whose dependence on the history is only through the cumulative cost. Since in our algos we consider MDPs whose state spaces are augmented by s_t, and from the definition of s_t, we can see that we are looking at the class of history-dependent policies. So, the feasibility assumption can be extended to policies over the augmented state (history-dependent policies). However, for simplicity and notational convenience, we use policies defined over the states (and not augmented states to become history-dependent) in our proposed PG algorithm.

"convergence proof of [27]"
There are currently two versions of [27] on arxiv, one dated April 15 that does not have a convergence proof, and one dated June 29 in which the authors provide a convergence proof. We clearly refer to the first version because it was the only one available before the NIPS deadline. The convergence proof in the 2nd version seems incomplete and lacks many necessary details. The authors use Thm. 5.2.1 in the book by Kushner & Yin and claim that the stochastic approximation converges to an asymptotically stable stationary point that is a local optimum. However, they do not verify or discuss whether all the assumptions of this theorem (A2.1 to A2.5) are satisfied. They neither provide a discussion on Lyapunov functions that are needed to prove a asymptotically stable fixed point, nor verify/discuss how the stochastic approximation visits the domain of attraction infinitely often with non-zero probability.

"experimental section"
Please see (a) above. Also a few points on the Tetris experiments in [27]: Since the CVaR metric in [27] is only valid for continuous random variables (Thm. 6.2 in the book by Shaprio), we are not sure whether it will be well-defined in a discrete problem like Tetris. Moreover, it is not clear how much risk aversion makes sense in Tetris.

Rev22

"extends [27] in several ways"
We believe that the algorithmic and theoretical contribution of this work is far beyond just extending the algorithm in [27]. What we do not cover here compared to [27] is the issue of importance sampling.

"comparison against Tamar"
Please see (b) above.

"compare to [27] with importance sampling (IS)"
Using IS to improve the sample efficiency of the CVaR optimization algos is an important issue that requires careful consideration. We never claimed in the paper that our AC algos are a replacement for using IS in CVaR optimization that the reviewer asks for comparison against "the approach of IS of Tamar". In fact, we list the appropriate use of IS in CVaR optimization as a future work in Sec. 7, because we believe that it requires more work as neither the approach of [27] nor of [2] (five years before [27]) completely solve this problem.

Rev23

"algos heuristic with no meaningful guarantees"
The reviewer doesn't mention anything to support this statement and it even contradicts his summary that "the paper addresses an important topic in a well-defined way". We can understand that converging to a local optimum might not be satisfactory, but this does not make the algorithms that are all mathematically founded and have convergence proof heuristic. This way any gradient-based method including many RL algos can be considered "heuristic with no meaningful guarantees".

"extension to other convex/coherent risk measures"
We agree with the reviewer that it would be nice to see how the machinery presented in our work can be extended to other convex/coherent risk measures. We consider this as an important future work.

"experiments are limited"
Please see (a) above. We totally agree with the reviewer that applying our algos to problems in finance and resource management that actually require risk aversion will significantly improves the quality of the paper. However, developing algos, proving their convergence, and applying them to a real-world problem might be beyond the scope of a conference paper.